# The origin of the odorant receptor gene family in insects

**Philipp Brand[1†]\*, Hugh M Robertson[2†]\*, Wei Lin[3], Ratnasri Pothula[4], William E Klingeman[5], Juan Luis Jurat-Fuentes[4], Brian R Johnson[3]**

[1]Department of Evolution and Ecology, Center for Population Biology, University of California, Davis, Davis, United States; [2]Department of Entomology, University of Illinois at Urbana-Champaign, Urbana, United States; [3]Department of Entomology and Nematology, University of California, Davis, Davis, United States; [4]Department of Entomology and Plant Pathology, University of Tennessee, Knoxville, United States; [5]Department of Plant Sciences, University of Tennessee, Knoxville, United States

**Abstract** The origin of the insect odorant receptor (OR) gene family has been hypothesized to have coincided with the evolution of terrestriality in insects. Missbach et al. (2014) suggested that ORs instead evolved with an ancestral OR co-receptor (Orco) after the origin of terrestriality and the OR/Orco system is an adaptation to winged flight in insects. We investigated genomes of the Collembola, Diplura, Archaeognatha, Zygentoma, Odonata, and Ephemeroptera, and find ORs present in all insect genomes but absent from lineages predating the evolution of insects. Orco is absent only in the ancestrally wingless insect lineage Archaeognatha. Our new genome sequence of the zygentoman firebrat *Thermobia domestica* reveals a full OR/Orco system. We conclude that ORs evolved before winged flight, perhaps as an adaptation to terrestriality, representing a key evolutionary novelty in the ancestor of all insects, and hence a molecular synapomorphy for the Class Insecta.
DOI: https://doi.org/10.7554/eLife.38340.001

**\*For correspondence:**
pbrand@ucdavis.edu (PB);
hughrobe@uiuc.edu (HMR)

[†]These authors contributed equally to this work

**Competing interests:** The authors declare that no competing interests exist.

## Introduction

From bacteria to mammals, living organisms of all levels of complexity have evolved chemosensory receptors to detect and discriminate chemicals in the environment (*Wuichet and Zhulin, 2010*; *Hansson and Stensmyr, 2011*). The largest metazoan gene families encode tens to hundreds of odorant receptors (ORs) that interact with volatile chemicals at the sensory periphery underlying the sense of smell (*Sánchez-Gracia et al., 2009*). OR gene families have evolved multiple times throughout the metazoans, including independent origins in vertebrates, nematodes, and insects (*Hansson and Stensmyr, 2011*). In insects, the OR gene family evolved from within the ancestral gustatory receptor (GR) gene family (*Scott et al., 2001*; *Robertson et al., 2003*) that extends back to ancient metazoan lineages (*Robertson, 2015*; *Saina et al., 2015*; *Eyun et al., 2017*). ORs are absent from non-insect arthropod genomes (*Peñalva-Arana et al., 2009*; *Almeida et al., 2015*; *Gulia-Nuss et al., 2016*; *Ngoc et al., 2016*; *Eyun et al., 2017*), and have been hypothesized to have evolved concomitant with the evolution of terrestriality in insects (*Robertson et al., 2003*).

The lack of molecular resources for ancestrally wingless (apterygote) insects and non-insect hexapods (*Figure 1*) has prevented the precise dating of the origin of insect ORs. Only recently, whole-genome sequencing efforts suggested that ORs are absent in non-insect hexapods such as Collembola (*Wu et al., 2017*) but present in derived winged (pterygote) insects such as damselflies (Odonata; *Ioannidis et al., 2017*). Efforts to understand more precisely the origin of the OR gene family within hexapods were greatly advanced by the findings of *Missbach et al. (2014)*, who sequenced

transcriptomes of the chemosensory organs of two apterygote insects, the bristletail *Lepismachilis y-signata* (Archaeognatha) and the firebrat *Thermobia domestica* (Zygentoma). They identified three ORs in the firebrat, which they named TdomOrco1-3, due to apparent similarity to the odorant receptor co-receptor (Orco; *Vosshall and Hansson, 2011*). Orco is a highly conserved single-copy gene present in all other insects studied to date and encodes a protein that is a partner with each of the other 'specific' ORs (*Benton et al., 2006*), a dimer required for OR-based olfaction in insects (*Larsson et al., 2004*). In contrast, *Missbach et al. (2014)* could not find ORs or Orco relatives in their bristletail transcriptome, instead finding only members of the ionotropic receptor (IR) gene family. Given evidence that IRs serve olfactory roles in terrestrial crustaceans and insects (*Rytz et al., 2013*; *Groh-Lunow et al., 2014*; *Rimal and Lee, 2018*), they argued that olfaction in terrestrial non-insect hexapods and apterygote insects is entirely IR-dependent, with Orco evolving as ancestral OR from the GR lineage between the Archaeognatha and Zygentoma. Based on these findings, *Missbach et al. (2014)* suggested that the Orco/OR system evolved together with flight in pterygote insects and left off with the observation that 'the existence of three Orco types remains mysterious'.

Recently, phylogenetic analysis of the OR gene family of the damselfly *Calopteryx splendens* suggested that at least one of the three Orco-like ORs from *T. domestica*, TdomOrco3, might be a specific OR instead of an Orco (*Ioannidis et al., 2017*). If this is correct, then the entire Orco/OR system evolved before winged insects, which would explain the 'mystery' of three apparent Orco types in Zygentoma. In an effort to identify the origin of the insect OR gene family and the Orco/OR system, we investigated the genome sequences of species belonging to multiple insect and other terrestrial hexapod orders (*Figure 1*), specifically Collembola (springtails), Diplura (two-pronged bristletails),

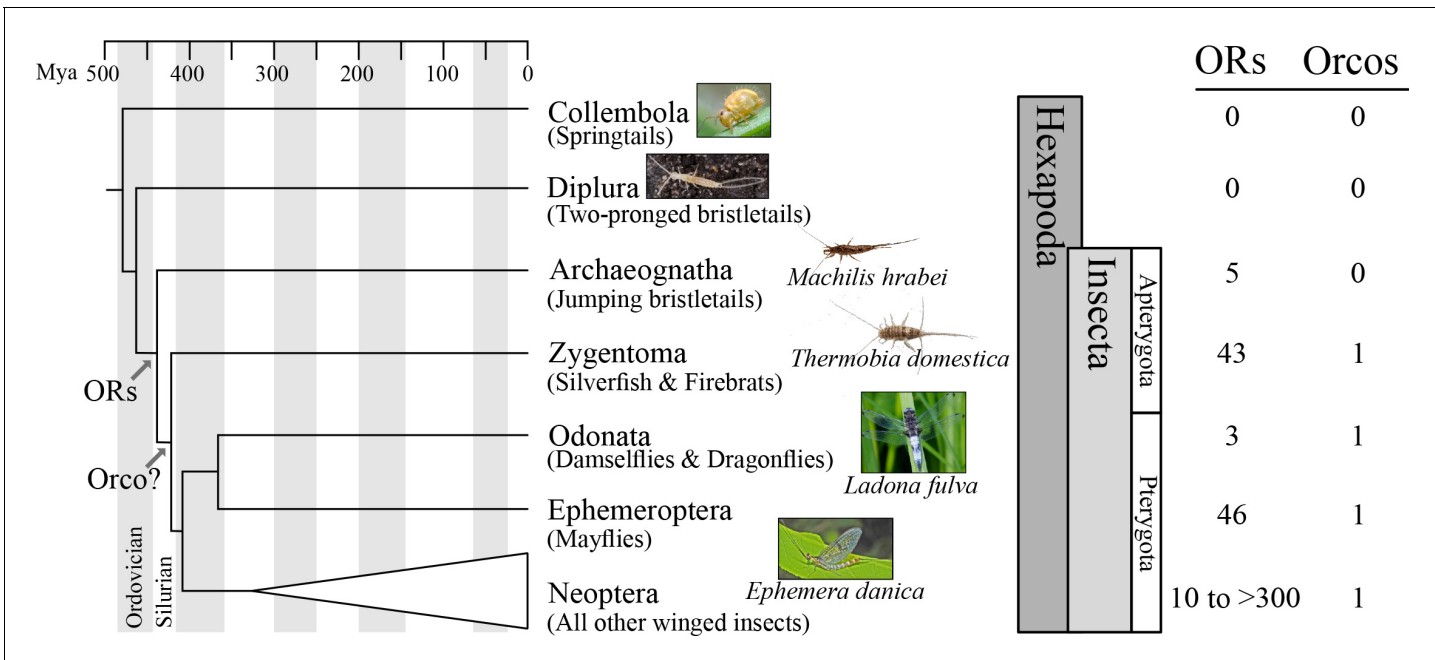

**Figure 1.** Origin of the insect odorant receptor gene family. The number of ORs and OR co-receptors (Orcos) for all species of the insect and other hexapod orders analyzed was mapped on the hexapod phylogeny sensu (*Misof et al., 2014*). ORs are present in all insects but absent from non-insect hexapod genomes, and thus likely represent an evolutionary novelty for the Class Insecta. Orco is present in all but Archaeognatha, an ancestrally wingless (apterygote) insect order. This suggests two scenarios including either the loss of Orco in Archaeognatha or an Orco origin following the evolution of ORs (as indicated). The genomes of all neopteran insects analyzed to date encode ORs, ranging from 10 ORs in head lice (*Kirkness et al., 2010*) to more than 300 ORs in ants (*Smith et al., 2011a*, *2011b*).
DOI: https://doi.org/10.7554/eLife.38340.002

Archaeognatha (jumping bristletails), Zygentoma (silverfish and firebrats), Odonata (damselflies and dragonflies), and Ephemeroptera (mayflies).

## Results and discussion

### ORs were present in the ancestor of insects

We detected no ORs in two non-insect hexapod lineages, Collembola and Diplura (a genome sequence is not available for the third lineage, the Protura), despite extensive annotation efforts. In contrast, we identified genes with similarity to known insect ORs in all insect genomes investigated (*Figure 1*; Tables S2 and S3 in *Supplementary file 1*). These included one species of each of the two apterygote insect orders, the Archeognatha and Zygentoma, as well as the pterygote orders Odonata and Ephemeroptera. Accordingly, ORs were likely present in the ancestor of all insects but absent from all non-insect hexapod lineages. This suggests that the origin of the OR gene family coincided with the evolution of insects. Thus, our analysis does not support the hypothesis that ORs evolved with the evolution of winged flight in insects (*Missbach et al., 2014*) but is compatible with the hypothesis that they evolved with terrestriality in insects (*Robertson et al., 2003*).

### The *Thermobia domestica* genome harbors a full Orco/OR gene family repertoire

With the exception of the Zygentoma, all orders analyzed had genome data either published (*Faddeeva-Vakhrusheva et al., 2016, 2017*; *Wu et al., 2017*) or available from the i5k Pilot Project from the Baylor College of Medicine at the i5k Workspace@NAL (*Poelchau et al., 2015*). To complete taxon sampling, we produced a draft genome assembly for *T. domestica* (Table S1 in *Supplementary file 1*; *Supplementary file 2*), enabling direct comparison to *Missbach et al. (2014)*. We find that the *T. domestica* genome encodes far more than the three Orco-like OR proteins they described. Our manual annotation revealed 43 ORs encoded by 32 genes including the three previously identified genes (TdomOrco1-3; *Missbach et al., 2014*). Four genes are modeled as exhibiting alternative splicing leading to the additional protein isoforms. We used the antennal transcriptome of *Missbach et al. (2014)* for support of intron-exon boundaries, however only a few transcriptome reads mapped to the 'specific' OR genes (Table S2 in *Supplementary file 1*), indicating that these ORs might be expressed in untested tissues or life stages, or at such low levels that the RNA-seq analysis of *Missbach et al. (2014)* did not sequence to a sufficient depth to recover these low-expressed transcripts.

Phylogenetic analyses of all ORs we annotated in the bristletail *Machilis hrabei* (5 ORs), the dragonfly *Ladona fulva* (4 ORs), and the mayfly *Ephemera danica* (47 ORs), as well as the previously annotated damselfly *C. splendens* (6 ORs; *Ioannidis et al., 2017*) revealed that one of the *T. domestica* ORs (TdomOrco2) clustered confidently with the Orco lineage in pterygote insects (*Figure 2*; *Supplementary file 3*). We believe this is the sole Orco relative because it shares unique features with the pterygote Orco proteins, such as a TKKQ motif in the expanded intracellular loop 2 (positions 327–330 in DmelOrco), and so we rename it simply TdomOrco. TdomOr1-8 are a set of Orco-like proteins that share a common gene structure with TdomOrco, with introns in phases 0-2-0-0-0. These last four introns are present in all the other TdomOr genes, as well as those of the bristletail, odonates, and mayfly, and correspond to the four introns identified by *Robertson et al. (2003)* as being ancestral to the OR family. The first phase-0 intron of Orco and Or1-8 is the only additional intron shared by most pterygote Orco genes.

With the exception of the bristletail *M. hrabei*, all insect genomes analyzed have both single genes with high similarity to Orco and multiple genes with similarity to specific ORs. The *M. hrabei* genome did not encode an Orco, but instead contains 5 ORs of high similarity that form a highly supported clade in the gene phylogeny (*Figure 2*). We also could not find an Orco in the deep RNA-seq transcriptome *Missbach et al. (2014)* generated for their bristletail, *L. y-signata*. While it is formally possible that *M. hrabei* has an Orco gene that is not present in the draft genome assembly due to an assembly gap, unsuccessful searches of raw reads from this genome project make this unlikely. We consider two possibilities. First, bristletails might have lost their Orco gene, however this loss must have occurred recently in both bristletail species because their 'specific' ORs are still intact. Second, the OR family might have originated with a few specific ORs like those of *M. hrabei*,

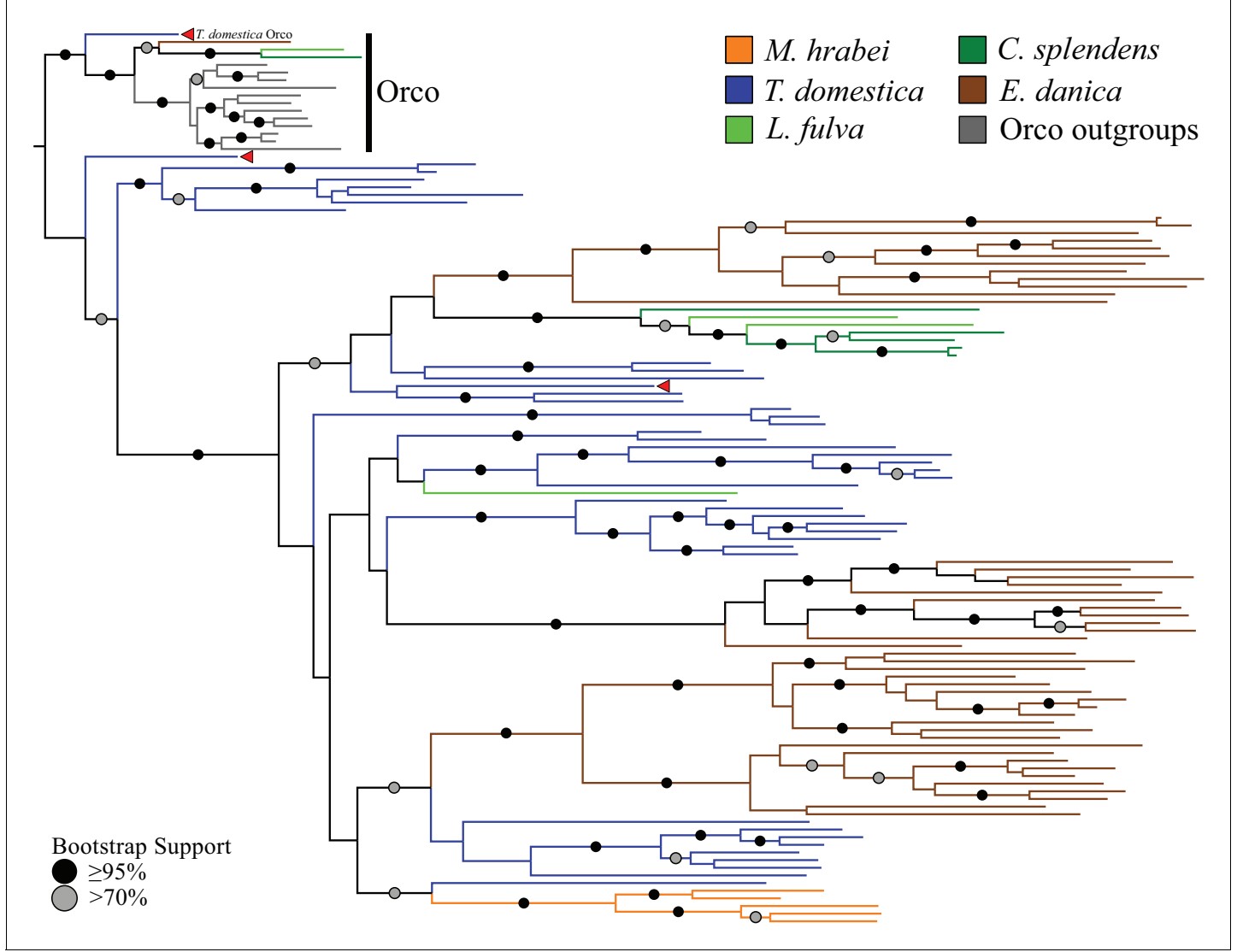

**Figure 2.** Odorant receptor (OR) gene family phylogeny including representatives of all apterygote and paleopteran insect orders. The Maximum Likelihood tree demonstrates monophyly of the single-copy insect Orco with high bootstrap support. The *M. hrabei* genome lacks Orco but encodes five ORs clustering in a single highly-supported clade. *T. domestica* has a fully developed functional OR/Orco system. The red arrowheads indicate the locations of the three *T. domestica* ORs identified by **Missbach et al. (2014)**, including the gene identified as *T. domestica* Orco in this study (formerly TdomOrco2).

DOI: https://doi.org/10.7554/eLife.38340.003

with the Orco lineage evolving between Archaegnatha and Zygentoma. Phylogenetic analysis using various sets of GRs from other insects, arthropods, and animals as outgroup to root the OR family tree does not resolve this question confidently (data not shown). In any case, if indeed functional, these five specific ORs in *M. hrabei*, at least one of which is present in *L. y-signata*, apparently function in the absence of Orco, perhaps alone or as dimers.

Finally, we note that while insects are defined by many morphological and developmental novelties, to the best of our knowledge the OR gene family is the first molecular novelty (i.e. synapomorphy) for the Class Insecta.

# Materials and methods

## Sequencing, assembly, and assessment of the *Thermobia Domestica* genome

Sequencing, assembly, and repetitive element annotation of the *T. domestica* genome followed *Brand et al. (2018)*. DNA was extracted from a single individual using a Qiagen DNeasy kit following the manufacturer's instructions. DNA libraries were constructed following the Truseq DNA PCR Free library preparation kit instructions for 250 bp paired-end sequencing on an Illumina HiSeq 2500 to produce a sequencing library required for DISCOVAR v1 (*Weisenfeld et al., 2014*) assembly. Assembly was performed using default parameters followed by Redundans v1 (*Pryszcz and Gabaldón, 2016*) to collapse contigs with ≥85% similarity in order to remove duplicates originating from expected high heterozygosity of outbred individuals. Finally, Agouti v1 (*Zhang et al., 2016*) was used with default parameters for scaffolding of the initial assembly. The quality and completeness of the assembly was assessed with standard N statistics and the Busco pipeline v2 (*Simão et al., 2015*) using the arthropoda_odb9 database (*Zdobnov et al., 2017*) in genome mode. In addition, a Jellyfish v2 (*Marçais and Kingsford, 2011*) produced k-mer frequency spectrum (k = 25) was used to assess the level of heterozygosity in the data and to estimate genome size on the basis of the consecutive length of all reads divided by the sequencing depth following *Brand et al. (2017)*.

The *T. domestica* genome assembly resulted in 618,474 scaffolds with an N50 of 15.5 Kb and a total length of 5.5 Gb. This exceeds the estimated genome size of 4.6 Gb by about 1 Gb. This could be due to either high heterozygosity of the sequenced individual, an imprecise genome-size estimate, or both. Since we were unable to produce inbred lines, it is possible that the genome of the sequenced individual was highly heterozygous. Indeed, our k-mer analysis revealed a high level of heterozygosity in the sequencing data, as indicated by the presence of a heterozygous peak of about twice the size of the homozygous peak (*Supplementaryfile 2*).

Despite the high level of heterozygosity in the sequencing data, BUSCO analysis suggested a low level of gene duplication. Only 7% of all 1066 benchmark genes tested were duplicated. In total, 90.8% of the BUSCO genes were present in the assembly, 63.6% of which were complete, and 27.2% incomplete. Although this suggests that the genome assembly is fragmented, it likely contains the majority of all genes at least in part. Indeed, the assembly was sufficient for the manual reconstruction of the odorant receptor gene set (described below).

Repetitive genome content was estimated based on transposable element (TE) prediction via the RepeatModeler – RepeatMasker pipeline (*Smit and Hubley, 2015*). TEs represent by far the largest fraction of repetitive elements in insect genomes, and thus provide an adequate estimate of overall repetitiveness of the genome (*Tribolium Genome Sequencing Consortium et al., 2008*; *Kapheim et al., 2015*, *Brand et al., 2017*, *Brand et al., 2018*). We first ran RepeatModeler for de novo repeat element family detection on the assembly followed by RepeatMasker for the annotation and classification of repetitive elements within the genome. Although this approach cannot detect lineage-specific repeat elements missing from the RepeatModeler database, it provides a good estimate of the overall level of genome repetitiveness (*Tarailo Graovac and Chen, 2002*; *Kapheim et al., 2015*; *Brand et al., 2017*).

The RepeatModeler analysis identified a total of 1812 repeat element families in the assembly, most of which could not be classified into known TE families (65.7%). We used the newly detected repeat elements as input database for RepeatMasker and annotated 9,986,807 individual elements with a cumulative length of 2.9 Gb, representing about half of the entire assembly length (52.9%). This suggests that the genome is highly repetitive. Of all elements, 4,330,119 (43.4%) could be classified into known TE families while the majority remained unclassified (Table S1 in *Supplementary file 1*). This is characteristic of non-model taxa that have not been previously studied on a molecular level, and thus not surprising since we are presenting the first comprehensive genetic resource of the insect order Zygentoma.

Overall, our assessment suggests that our approach produced a genomic resource of sufficient quality for the inference of gene family evolution. The BUSCO analysis indicates that most genes are present in the assembly. However, low N50 and a moderate level of incomplete BUSCO benchmarking genes indicates assembly fragmentation. Fragmentation is likely influenced by the high level of repetitive elements (>50%) in the genome which could interfere with the DISCOVAR approach that

is based on a single library type of small insert size (*Weisenfeld et al., 2014*). While DISCOVAR assembly was reported to perform well on some insect genomes of comparatively small size (*Love et al., 2016*), our results suggest that a high fraction of repetitive elements in the genome data can negatively influence assembly performance.

In addition to a moderate level of fragmentation, duplication levels of homologous genomic regions are likely slightly elevated in the assembly, especially for regions of high heterozygosity. The observed discrepancy between genome size estimate and assembly length suggests that our Redundans approach could not entirely remove homologous genomic regions of high heterozygosity (i.e. duplicates) in the DISCOVAR assembly, despite conservative settings. It is thus important to adjust gene annotation efforts accordingly (described below).

While a higher level of contiguity and thus assembly quality simplifies gene annotation and genome-level analyses, genomes of similar quality to the one we present have been successfully used to study genome evolution (e.g. some genomes in *Kapheim et al., 2015*; *Neafsey et al., 2015*). Especially for non-model organisms with high heterozygosity and large genome sizes like *T. domestica*, producing high-quality genomes comparable to model organisms such as *Drosophila melanogaster* remains a non-trivial task (*Richards, 2018*). The genome assembly in this study represents the first Zygentoma genome resource and thus will be highly useful in future studies of insect genome evolution.

## Odorant receptor annotation

The genomes of the dipluran *Catajapyx aquinolaris*, the collembolans *Holacanthella duospinosa*, *Orchesella cincta*, and *Folsomia candida*, the firebrat *Thermobia domestica*, the bristletail *Machilis hrabei*, the dragonfly *Ladona fulva*, and the mayfly *Ephemera danica* were used for manual OR gene annotation.

### The firebrat *Thermobia Domestica*

The *T. domestica* assembly v1 described above was searched for ORs and models for them were manually built in a text editor (TextWrangler, which allows up to 18 kb of DNA sequence on each line). Protein translations were aligned against each candidate exonic region and exon-intron boundaries defined by a combination of support from RNAseq (see below), the shared intron locations and phases across genes (see below), and predictions using the Splice Site Prediction by Neural Network server at the Berkeley Drosophila Genome Project (http://www.fruitfly.org/seq_tools/splice.html).

A variety of strategies were employed to find and build models for 32 OR genes. Initially TdomOrco1-3 from *Missbach et al. (2014)* were used as queries to search using TBLASTN, which revealed that these three genes, for which *Missbach et al. (2014)* provided full-length cDNAs, are split across 7, 4, and 3 scaffolds, respectively (now renamed Orco, Or1, and Or9 in Table S2 in *Supplementary file 1*). These genes are revealed to be rather large, with long introns of considerable, and sometimes unknown, length as they include gaps of unknown length between scaffolds, explaining the large size of the genome assembly. Thus, Orco is at least 55 kb long, Or1 is at least 14 kb long, and Or9 is at least 38 kb long. These searches also revealed that our genome assembly is often redundant for two OR haplotypes, presumably resulting from the majority of the sequence being generated from a single diploid individual of high heterozygosity (described above). That these three genes were almost completely represented in the genome assembly, often in two very similar copies, with only 9 bp missing from Orco, gave us confidence that the assembly contains most unique sequence from the genome, and that we might therefore be able to find and build complete models for additional OR genes.

These three TBLASTN searches were performed with sensitive settings to detect even distantly related genes, and revealed many additional related genes, however like the first three, many exons were in separate scaffolds (Table S2 in *Supplementary file 1*). These sensitive searches used an E value of 1000 and word size of 2 with no filter to allow us to find divergent and sometimes short exons. We used seven strategies to assist in confidently and completely building models for almost all of these genes. First, some genes were clearly contained within single scaffolds, so models for them could be confidently built given some RNAseq support (see below), the shared gene structures of these genes (see below), and their sequence similarity. These genes are Or2, 3, 4, 18, 19, 21, 23, 24a/b, and 26. Their sizes of 20, 16, 24, 14, 43, 35, 35/29, and 14 kb, respectively, confirm the large

typical size for these OR genes in this large genome. Iterative TBLASTN searches using the sensitive settings described above with these proteins and others confidently built below, allowed nearly complete annotation of the OR family in this genome assembly.

Second, we employed the antennal RNAseq information from *Missbach et al. (2014)*, and to a lesser extent the 1Kite project that sequenced from whole animals hence has far less representation for genes usually expressed in antennae (*Misof et al., 2014*), to establish or confirm gene models as well as connections between exons on different scaffolds. These searches were done using MEGA-BLAST of the raw reads at the SRA, with word size of 16. This kind of experimental evidence is particularly valuable and sometimes allowed us to connect exons on different scaffolds that we might otherwise not have been able to associate confidently. Introns supported by spanning RNAseq reads (30 in total), as well as the three cDNAs from *Missbach et al. (2014)*, are indicated in bold type in Table S2 in *Supplementaryfile 1*.

Third, for this report we focused on *T. domestica* because of the availability of the above RNAseq information as experimental support, however we also generated a genome assembly of similar contiguity and haplotype redundancy for another zygentoman, the silverfish *Ctenolepisma longicaudata*. This genome sequence will be reported elsewhere, however we employed it here to assist in connecting exons on different scaffolds in *T. domestica*. We were able to build complete or nearly complete models for many apparently orthologous genes in *C. longicaudata* in single scaffolds, which by comparison then allowed confident connections of exons on different scaffolds in *T. domestica*. Even adjacent pairs of exons in single scaffolds in *C. longicaudata* sometimes allowed connections in *T. domestica* that facilitated completion of gene models. The resultant apparently orthologous proteins shared 50–88% amino acid identity between these two species (shown for most simple orthologous genes in Table S2 in *Supplementary file 1*). All gene models that employed one or more such connection are indicated with an asterisk after the suffix J after their names in Table S2 in *Supplementary file 1*.

Fourth, the redundancy of these two genome assemblies with two haplotypes for many genes occasionally assisted in connecting exons that were separately connected to other flanking exons in separate scaffolds, presumably representing the two haplotypes in each assembly. The scaffolds listed in Table S2 in *Supplementary file 1* are generally the longest scaffolds containing one or more exons.

Fifth, for a few recently-duplicated genes like Or5-8, we associated the front and back halves on the basis of the availability of appropriate combinations of exons within scaffolds, and similarity to a single relative in *C. longicaudata*. It is possible that these models are not completely correct, however, as they constitute a small *Thermobia*-specific expansion, any such errors will not affect our phylogenetic analysis. Nevertheless, the final short exon could not be confidently identified for Or7.

Sixth, these genes all contain four short exons encoding the C-terminal regions of their proteins, and these are separated from the N-terminal-encoding exon(s) by a phase-2 intron followed by three phase-0 introns. To make TBLASTN searches for these short exons even more sensitive, particularly for divergent genes, two additional 'amino acids' were added to the ends of the query sequence encoded by the most closely related confident gene model, specifically FR to represent a phase-2 consensus intron acceptor, VS to represent a phase-0 donor, and LQ to represent a phase-0 acceptor.

Seventh, four gene models have parts missing from the assembly. Orco has 9 bp missing from the ends of two scaffolds that contain the front and back halves of exon1, while Or25 has ~250 bp missing from within exon1 for similar reasons. In the case of Orco these 9 bp were provided by the cDNA from *Missbach et al. (2014)* as well as raw genome reads in the SRA, and the latter were used to repair the assembly for Or25. As noted above, the final exon for Or7 could not be confidently identified, and the same is true for Or16.

Four of these genes are modeled as being alternatively-spliced in a fashion seen commonly in insect genomes, including the original descriptions of this family from *D. melanogaster* (*Clyne et al., 1999*; *Vosshall et al., 1999*). This alternative splicing is somewhat unusual because it involves multiple long first exons in tandem array that are alternatively spliced into the four short exons encoding the C-terminus. These genes are Or15a-c, 24a/b, 27a-d, and Or31a-f (Table S2 in *Supplementary file 1*). In the case of Or15, the three long first exons plus exon2 are contained within one scaffold, so this alternative splicing, while not supported by RNAseq reads between these three first exons and exon2, seems certain. Or24a/b are completely contained within one scaffold,

and while again there are no RNAseq reads to support the alternative-splicing model, it appears confident. The Or27a-d locus is more complicated, because the Or27a first exon is in a separate scaffold, while the Or27b/c exons are in another, the Or27d plus exons2/3 in yet another, and the last two short exons in a fourth scaffold. Nevertheless, we believe this model because there is an RNAseq read that connects the Or27a first exon to the shared exon2, which along with exon3 is in the same scaffold as the Or27d first exon. The first exons for Or27b/c are together in a separate scaffold, but are so similar in sequence to Or27a (75% and 71% amino acid identity) that they most likely are also alternatively spliced, and exons4/5 in the fourth scaffold are most similar to the related Or25/26 genes. Finally, Or31a-f is also a complicated locus spread across five scaffolds with no RNA-seq support for the alternative splicing model, which we nevertheless believe is confident because a comparable alternatively-spliced locus exists in *C. longicaudata*.

Our ability to construct models for 32 OR genes encoding 43 proteins in this *T. domestica* genome assembly confirms our initial observation that the assembly contains most unique sequence from the genome. The only possible additional protein isoform we excluded is that there is a first exon 17 kb upstream of Orco in scaffold Ther_dom_306089 that could conceivably be alternatively spliced into Orco, yielding two Orco isoforms with only 31% amino acid identity for their first 200 amino acids. The same kind of arrangement is found in *C. longicaudata*, suggesting that it is a conserved and functional arrangement. Nevertheless, without RNAseq support for this model from *Missbach et al. (2014)* and the complete absence of alternative splicing for the single Orco gene in all other described insects, we chose not to include this potential divergent isoform.

Two of the alternatively-spliced isoforms are apparent pseudogenes (Or15c and Or24b), each with single stop codons in the first long exon, but it is possible that these are pseudo-pseudogenes, as shown for some OR genes in *Drosophila* flies (*Prieto-Godino et al., 2016*), otherwise the remaining 41 proteins appear to be functional.

## The dragonfly *Ladona fulva* and the mayfly *Ephemera danica*

These two draft genome assemblies were accessed at the i5k Workspace@NAL (*Poelchau et al., 2015*) where they are presented from the i5k Pilot Project at the Human Genome Sequencing Center at the Baylor College of Medicine. The dragonfly and mayfly OR families were manually annotated using the Apollo genome browser. These gene models and encoded proteins will eventually be available from these genome projects, and the proteins are provided in the supplement.

The four OR models including Orco for *L. fulva* are full-length and intact. For *E. danica*, in addition to Orco there are 46 OR models, 7 of which are partial models with termini or internal regions missing due to gaps in the assembly that could not be repaired with raw genome reads from the Sequence Read Archive (SRA) at the National Center for Biotechnology Information (NCBI), while 10 are pseudogenes.

## The bristletail *Machilis hrabei:*

The bristletail genome from the i5k pilot project is not available in the Apollo browser, so models for the five OR genes in that draft genome sequence were manually built in a text editor. All five required considerable repairs of both ambiguous bases and assembly gaps using raw genome reads from the SRA, and three were joined across scaffolds (Table S3 in *Supplementaryfile 1*). There is considerable RNAseq support for four of these five models from the whole-body RNAseq of the i5k and 1Kite projects (found with MEGABLAST searches at the SRA at NCBI), as well as five matching reads for MhraOr1 from the RNAseq of the related species employed by *Missbach et al. (2014)*, *Lepismachilis y-signata*, detected using TBLASTN searches of raw reads downloaded from the SRA (Table S3 in *Supplementary file 1*). A first and second exon in 18 kb scaffold_47427 that encode 95% identical amino acids to MhraOr4 were not included in the analysis as they are a partial model and do not contribute significantly to the diversity of MhraOrs.

We note that the first two pairs of these MhraOr genes are in inverse orientation to each other, with their tails adjacent (Table S3 in *Supplementary file 1*), a feature also seen with some of the TdomOrs, although they also have some genes in the head-to-head orientation, for example Orco/Or5, Or10/25, Or11/12/22, and Or13/14 (Table S2 in *Supplementary file 1*).

We made extensive efforts to identify Orco in *M. hrabei* and *L. y-signata* using TdomOrco as query. There are no obvious matches for any part of Orco in the genome assembly of *M. hrabei*

using TBLASTN searches. Because this is a highly conserved protein (for example, mostly co-linear and 88/45/42% identical between *T. domestica* and *C. longicaudata*, *E. danica*, and *L. fulva*, respectively), it should be readily discovered. It is always possible that a gene might be completely unassembled in a draft genome assembly, although unlikely given we were able to assemble full-length models for MhraOr1-5, so we undertook a TBLASTN search of a set of raw genome reads that gave 5-10X coverage for MhraOr1-5 and found nothing. We also did TBLASTN searches of the raw whole-body RNAseq reads from both the i5k *M. hrabei* project and the 1Kite set without success. Finally, in other insects Orco is generally well expressed in antennae and in TBLASTN searches of the raw reads from the antennal RNAseq of *Missbach et al. (2014)* for *L. y-signata* we found several reads that encode an Orco protein and were able to assemble a 713 bp contig encoding 237 amino acids, however it is a contaminant with matches of 85% identity to various lepidopteran Orco proteins. We therefore conclude that these two bristletails do not have an Orco gene. It is also noteworthy that the five MhraOr genes have the same gene structure as the TdomOr9-31 genes, without the additional phase-0 intron splitting the long first exon seen in TdomOrco and the Orco-like TdomOr1-8 (Table S3 in *Supplementary file 1*), so there is no hint of Orco-like genes in this bristletail genome.

### The basal hexapod lineages collembola and diplura

We similarly exhaustively examined the draft genome assemblies for three Collembola (*Faddeeva-Vakhrusheva et al., 2016*, *2017*; *Wu et al., 2017*) as well as a dipluran from the i5k pilot project, *Catajapyx aquinolaris*, and found no evidence of either Orco or ORs, in agreement with *Wu et al. (2017)*.

## Phylogenetic analysis

OR protein alignments were produced with CLUSTALX v2 (*Larkin et al., 2007*) and gaps were trimmed using Gappyout in TrimAl (*Capella-Gutiérrez et al., 2009*). The resulting alignment left 434 characters including a variable N-terminus caused by length variation in the first and second intracellular loop. The resulting alignments were used for gene tree inference using RaxML (*Stamatakis et al., 2005*) under the JTT + G substitution model which previously has been found to be the best model for OR gene trees (*Brand and Ramírez, 2017*). We used a total of 20 independent ML searches and 1000 bootstrap replicates (*Supplementary file 3*).

## Acknowledgements

We thank Leslie Vosshall, Roman Arguello, and one anonymous reviewer for helpful comments that improved the manuscript. We thank Stephen Richards for permission to examine unpublished genome sequences from the i5k pilot project, and Kimberly Walden for assistance with BLAST searches. This work was funded by a National Science Foundation grant, IOS-1456678, to Brian Johnson and Juan Luis Jurat-Fuentes, and a USDA Hatch grant to Brian Johnson (CA-D-ENM 2161-H).

## Additional information

### Funding

| Funder | Grant reference number | Author |
| --- | --- | --- |
| National Science Foundation | IOS-1456678 | Juan Luis Jurat-Fuentes Brian R Johnson |
| U.S. Department of Agriculture | Hatch CA-D-ENM 2161-H | Brian R Johnson |

The funders had no role in study design, data collection and interpretation, or the decision to submit the work for publication.

### Author contributions

Philipp Brand, Hugh M Robertson, Conceptualization, Data curation, Formal analysis, Methodology, Writing—original draft, Writing—review and editing; Wei Lin, Ratnasri Pothula, William E Klingeman,

Methodology, Writing—review and editing; Juan Luis Jurat-Fuentes, Funding acquisition, Methodology, Writing—review and editing; Brian R Johnson, Conceptualization, Funding acquisition, Writing—review and editing

### Author ORCIDs
Philipp Brand ⓘ http://orcid.org/0000-0003-4287-4753
Hugh M Robertson ⓘ http://orcid.org/0000-0001-8093-0950

### Decision letter and Author response
Decision letter https://doi.org/10.7554/eLife.38340.012
Author response https://doi.org/10.7554/eLife.38340.013

## Additional files
### Supplementary files
• Supplementary file 1. Table S1 Transposable element repeat class analysis of the *Thermobia domestica* genome assembly. Table S2 Details of the *Thermobia domestica* OR family genes and proteins. Columns are: Gene – the gene and protein name we are assigning (suffixes, which are not part of the name but indicate features of the gene model, are C – C-terminus missing, F – assembly was repaired, J – gene model spans scaffolds, * - one or more join across scaffolds made on the basis of comparison with an ortholog in *Ctenolepisma longicaudata* or a close intact relative in *Thermobia*); Scaffold – the v1 genome assembly scaffold ID; Coordinates – the nucleotide range from the first position of the start codon to the last position of the stop codon in the contig/scaffold; Strand – + is forward and - is reverse; RNA – number of independent pairs of reads from *Missbach et al. (2014)* and 1Kite (*Misof et al., 2014*); Introns – phases of introns (bold indicates those supported by Missbach et al. cDNAs, or their raw RNAseq reads, or those from 1Kite); % - percent identity for most apparent 1–1 orthologs with *Ctenolepisma longicaudata*; AAs – number of encoded amino acids in the protein; Comments – comments on the gene model. Note that Orco is Orco2, Or1 is Orco1, and Or9 is Orco3 of *Missbach et al. (2014)*. Table S3 Details of the MhraOr family genes and proteins. Columns are: Gene – the gene and protein name we are assigning (suffixes, which are not part of the name but indicate features of the gene model, are F – assembly was repaired, J – gene model spans scaffolds, * - one or more joins across scaffolds is based only on sequence similarity to the other proteins); Scaffold – the v1 genome assembly scaffold ID from i5k; Coordinates – the nucleotide range from the first position of the start codon to the last position of the stop codon; Strand –+is forward and - is reverse; RNA – number of independent pairs of reads from 1Kite and the i5k pilot project (single reads from *Missbach et al. (2014)* for the related species *Lepismachilis y-signata* are shown in parentheses); Introns – phases of introns (bold indicates those supported by RNAseq reads); AAs – number of encoded amino acids in the protein; Comments – comments on the gene model. FASTA format proteins for the newly described ORs: Suffixes, which are not part of the gene/protein name but indicate features of the gene model, are C – C-terminus missing, F – assembly was repaired, I – internal regions missing, J – gene model spans scaffolds, P – pseudogene. All proteins of the newly described ORs and the alignment used to reconstruct the gene tree are available on Dryad.
DOI: https://doi.org/10.7554/eLife.38340.004

• Supplementary file 2. kmer frequency spectrum of *Thermobia domestica* sequencing reads. A high heterozygous peak with a maximum at k = 17 in comparison to the homozygous peak around k = 37 indicates high heterozygosity in the genomic data. High heterozygosity is a known culprit to difficult genome assembly but cannot be avoided in most non-model organisms, which often cannot be used to produce inbred lines.
DOI: https://doi.org/10.7554/eLife.38340.005

• Supplementary file 3 Odorant receptor gene family phylogeny of all apterygote and paleopteran insect orders. The Maximum Likelihood phylogeny shows relationships between the ORs and Orcos detected in *M. hrabei* (orange), *T. domestica* (blue), *L. fulva* (bright green), *C. splendens* (dark green; *Ioannidis et al., 2017*), and *E. danica* (brown).
DOI: https://doi.org/10.7554/eLife.38340.006

• Transparent reporting form
DOI: https://doi.org/10.7554/eLife.38340.007

## Data availability

Raw genome sequence reads are being submitted to the Sequence Read Archive at the NCBI. The Thermobia domestica genome assembly is available from Dryad under doi: 10.5061/dryad.p2t8170

The following dataset was generated:

| Author(s) | Year | Dataset title | Dataset URL | Database, license, and accessibility information |
|---|---|---|---|---|
| Brand P | 2018 | Thermobia domestica genome assembly v 1.0 | http://dx.doi.org/10.5061/dryad.p2t8170 | Available at Dryad Digital Repository under a CC0 Public Domain Dedication |

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
