## [Decision Letter]

Thank you for submitting your article "The origin of the odorant receptor gene family in insects" for consideration by *eLife*. Your article has been reviewed by three peer reviewers, and the evaluation has been overseen by a Reviewing Editor and Diethard Tautz as the Senior Editor. The following individuals involved in review of your submission have agreed to reveal their identity: Leslie B Vosshall (Reviewer #2); Roman Arguello (Reviewer #3).

The reviewers have discussed the reviews with one another and the Reviewing Editor has drafted this decision to help you prepare a revised submission.

Summary:

This manuscript describes an interesting study of the evolutionary origins of the OR genes in insects. It is in "Research Advance" format, and is based on an earlier *eLife* article by Missbach et al. from 2014. Brand et al. consider the same issue as Missbach et al. but analyze a broader and deeper dataset, and arrive at some different conclusions. Missbach et al. suggested that the OR/Orco system is an adaptation to winged flight; Brand et al. provide convincing evidence that it evolved before winged flight. A species that Missbach et al. described as having 3 Orco-like genes and no ORs is found by Brand et al. to have one Orco and many ORs.

Essential revisions:

1) The paper is not written for a general readership: the authors should 'translate' the paper into language that is accessible to *eLife* readers (e.g., clarify the meaning of basal hexapods, synapomorphy, apterygote, pterygote etc.). To make the figures of greater visual interest and to help non-experts appreciate the profundity of the results, please include a representative photograph/graphic of each of the species analyzed. In particular, Figure 1 should be expanded by listing the species that are analyzed and their common names, as this would aid understanding of the text in which individual organisms are sometimes referred to by their species name, sometimes by the order in which they are classified, and sometimes by their common name.

2) The authors should indicate the caveat that a missing Orco gene in Archaeognatha might be a technical issue (i.e., a genome sequence gap) at the appropriate passages in the manuscript. It would be helpful if the authors include a summary table of the genome data coverage for the various species, and discuss further the different possible reasons for failure of Missbach et al. to find *T. domestica* ORs beyond the possibility of low sequence coverage in their RNA-seq datasets (e.g., these genes could be expressed in different tissues or life stages form that analysed).

3) Parsable alignment files should be included and the fasta sequences provided at the end of the Supplementary file 1 should be provided as a separate parsable file.

---

## [Author Response]

Essential revisions:1) The paper is not written for a general readership: the authors should 'translate' the paper into language that is accessible to eLife readers (e.g., clarify the meaning of basal hexapods, synapomorphy, apterygote, pterygote etc.). To make the figures of greater visual interest and to help non-experts appreciate the profundity of the results, please include a representative photograph/graphic of each of the species analyzed. In particular, Figure 1 should be expanded by listing the species that are analyzed and their common names, as this would aid understanding of the text in which individual organisms are sometimes referred to by their species name, sometimes by the order in which they are classified, and sometimes by their common name.

To make the manuscript more appealing and readable to a general readership we clarified specific terms such as *apterygote* and replaced jargon whenever possible. We further removed problematic and incorrectly used terms such as ‘basal hexapods’ and ‘ancient lineages’. To make the manuscript visually more appealing, we modified Figure 1. It now includes the common names for all orders analyzed, as well as pictures and full names of all four insect species newly annotated for ORs.

2) The authors should indicate the caveat that a missing Orco gene in Archaeognatha might be a technical issue (i.e., a genome sequence gap) at the appropriate passages in the manuscript. It would be helpful if the authors include a summary table of the genome data coverage for the various species, and discuss further the different possible reasons for failure of Missbach et al. to find T. domestica ORs beyond the possibility of low sequence coverage in their RNA-seq datasets (e.g., these genes could be expressed in different tissues or life stages form that analysed).

We modified the manuscript to address possible reasons for a missing Orco in Archeognatha more specifically, including the possibility of technical issues (–subsection “Sequencing, assembly, and assessment of the *Thermobia domestica* Genome”, fourth paragraph). We also changed the text to address the low number of OR genes reconstructed by Missbach et al. in more depth (subsection “The firebrat *Thermobia domestica*”, second paragraph). It is indeed very likely that not all ORs we detected in the genome are expressed in the life stage and tissue analyzed by Missbach et al. In addition, both of these points are now more deeply discussed in the text. We believe that the modifications to the text are now sufficient to provide a comprehensive discussion of all possible factors that might lead to a missing gene in a genome and an appropriate assessment of those factors with respect to our annotation efforts. A table reporting the sequencing depth of the genome would not add much to this discussion so we decided not to include one in the manuscript.

3) Parsable alignment files should be included and the fasta sequences provided at the end of the Supplementary file 1 should be provided as a separate parsable file.

All requested files were uploaded to Dryad.